There are amendments to this paper

# Deglacial water-table decline in Southern California recorded by noble gas isotopes

Alan M. Seltzer [1]*, Jessica Ng[1], Wesley R. Danskin [2], Justin T. Kulongoski[1,2], Riley S. Gannon [2], Martin Stute[3,4] & Jeffrey P. Severinghaus[1]

Constraining the magnitude of past hydrological change may improve understanding and predictions of future shifts in water availability. Here we demonstrate that water-table depth, a sensitive indicator of hydroclimate, can be quantitatively reconstructed using Kr and Xe isotopes in groundwater. We present the first-ever measurements of these dissolved noble gas isotopes in groundwater at high precision ($\leq 0.005‰$ amu$^{-1}$; $1\sigma$), which reveal depth-proportional signals set by gravitational settling in soil air at the time of recharge. Analyses of California groundwater successfully reproduce modern groundwater levels and indicate a $17.9 \pm 1.3$ m ($\pm 1$ SE) decline in water-table depth in Southern California during the last deglaciation. This hydroclimatic transition from the wetter glacial period to more arid Holocene accompanies a surface warming of $6.2 \pm 0.6$ °C ($\pm 1$ SE). This new hydroclimate proxy builds upon an existing paleo-temperature application of noble gases and may identify regions prone to future hydrological change.

[1] Scripps Institution of Oceanography, University of California, San Diego, 9500 Gilman Drive, La Jolla, CA 92093, USA. [2] California Water Science Center, United States Geological Survey, 4165 Spruance Road, San Diego, CA 92101, USA. [3] Lamont-Doherty Earth Observatory of Columbia University, 61 Route 9W, Palisades, NY 10964, USA. [4] Barnard College, 3009 Broadway, New York, NY 10027, USA. *email: aseltzer@ucsd.edu

Noble gases dissolved in groundwater have a wide range of physical applications for climatology and hydrogeology, owing to their chemical and biological inertness. For example, past mean-annual surface temperatures derived from noble gases in paleo-groundwater comprise some of the most reliable terrestrial temperature reconstructions of the last glacial period[1,2]. Physical models of groundwater recharge, transport, contamination, and age also are frequently constrained by noble gas measurements[3–5]. Although dissolved noble gas concentrations and helium isotopes are routinely measured, groundwater Kr and Xe stable isotope studies are rare because fractionation signals are generally smaller than typical order 1–10‰ amu$^{-1}$ analytical uncertainties.

Here we apply a newly developed technique[6] to make the first-ever measurements of stable Kr and Xe isotope ratios in groundwater at high precision (≤5 per meg amu$^{-1}$/1σ; where 1 per meg = 0.001‰ = 0.0001%). Measurement of Kr and Xe isotope ratios, along with Ar isotopes and Ar, Kr, and Xe concentrations, in 58 groundwater samples from 36 wells allowed us to test a simple fractionation model constrained by recent experimental determinations of noble gas isotopic solubility and diffusivity ratios[6]. Our findings suggest that Kr and Xe isotope ratios in groundwater record the depth to water at the time and place of recharge. We present an inverse model to reconstruct past water-table depth (WTD) based on noble gas measurements and demonstrate its accuracy in reproducing observed water levels in modern groundwater. We then apply it to a suite of paleo-groundwater samples from San Diego, California, which indicate a ~20-m decline in reconstructed WTD during the last deglaciation. We interpret this shift as a regional groundwater response to a major hydroclimatic change from the wetter glacial period to more arid Holocene. This finding is consistent with previous paleo-hydrological evidence and climate model simulations from the southwestern United States[7–11].

## Results and discussion

**Depth-dependent gravitational isotopic signals.** A theoretical model for gas-phase isotopic fractionation in porous media, validated by soil air observations, suggests that dissolved Kr and Xe stable isotope ratios in unsaturated-zone (UZ) air are primarily fractionated from the well-mixed atmosphere by gravitational settling[12,13]. Gravitational settling is a well understood process by which molecular diffusion in stagnant air in hydrostatic balance leads to a nearly linear increase in heavy-to-light gas ratios with depth[14]. Past measurements of inert gases in polar firn[15,16] and UZ air[12,13] have been found to be in close agreement with the theoretical gravitational settling slope (e.g. ~4.0 per meg amu$^{-1}$ m$^{-1}$ at ~20 °C).

Thermal diffusion[17] and steady-state UZ-to-atmosphere water-vapor fluxes[12] both weakly oppose the influence of gravitational settling on UZ air isotopic composition (Fig. 1). In the presence of a temperature gradient, thermal diffusion acts to concentrate heavy gases towards the cooler end of a column of gas. Thermal diffusion decreases heavy-to-light isotope ratios in deep UZ air, which is warmed by geothermal heat from below, relative to the atmosphere. Isotope ratios of the lighter noble gases (He, Ne, and Ar) are more strongly affected by thermal diffusion than Kr and Xe isotope ratios, per mass unit difference[18,19]. At steady-state, the flux of water vapor from the moist UZ to drier overlying surface air leads to a kinetic isotopic fractionation that also lowers heavy-to-light isotope ratios and affects the lighter noble gases more than Kr and Xe[12,13]. This fractionation is induced by vertical partial pressure gradients of dry air constituents between the atmosphere and UZ, which drive steady-state molecular

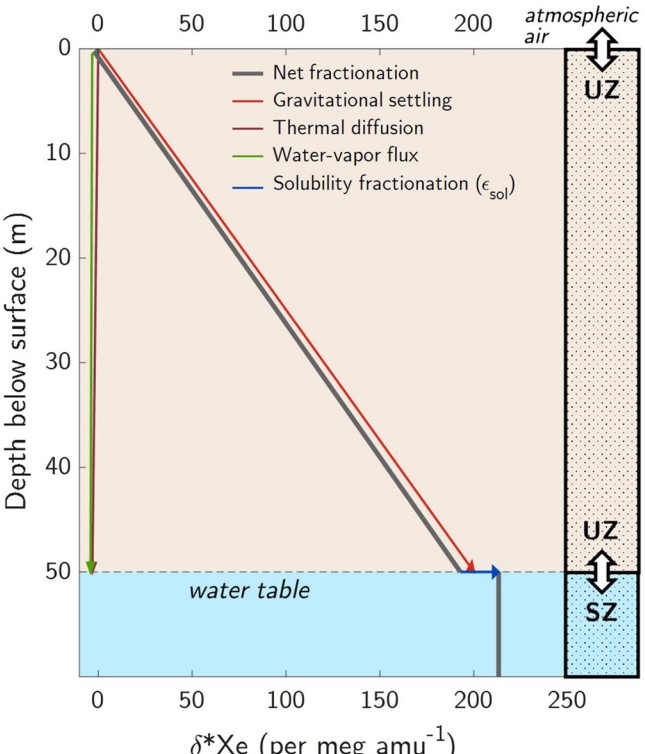

**Fig. 1 Steady-state xenon isotopic fractionation in an unconfined aquifer.** Gravitational settling causes mass difference-normalized, heavy-to-light xenon isotope ratios (δ*Xe) in unsaturated zone (UZ) air to increase with depth below the land surface, relative to atmospheric air. At the water table, dissolved gases in the saturated zone (SZ) equilibrate with deep UZ air. Solubility fractionation causes δ*Xe to further increase in the dissolved phase relative to the gas phase. Note that nearly all water-vapor flux fractionation occurs between the surface and shallow wetting front (at one meter depth in this idealized illustration), below which water-vapor flux fractionation is nearly constant with depth and only changes due to its weak sensitivity to temperature[12,13].

diffusion of atmospheric noble gases into the UZ against the upwards flux of water vapor[12].

Air at the bottom of the UZ dissolves into groundwater at the water table, thereby transferring the signal of UZ air fractionation to dissolved isotope ratios in groundwater, as depicted in an idealized model of an unconfined aquifer system (Fig. 1). Small isotopic solubility differences, constrained by recent determinations[6], lead to slight further increases of heavy-to-light isotope ratios. Over time, as the uppermost groundwater is gradually displaced downward by subsequent recharge, it is sequestered from overlying UZ air, prohibiting further gas exchange. After the time of recharge, which is here defined to be the last period of contact between a groundwater parcel and overlying UZ air, the dissolved noble gas composition is retained except for dispersive intra-aquifer mixing. Therefore, the noble gas composition of old groundwater reflects conditions set at the water table at the time and place of recharge[20,21].

In this conceptual framework, a dissolved isotope ratio ($\delta_{diss}$) is cumulatively fractionated at the time of recharge from its atmospheric ratio ($\delta_{atm}$, where $\delta_{atm} = 0$ by definition) by steady-state diffusive processes in UZ air ($\varepsilon_{UZ}$), isotopic solubility fractionation ($\varepsilon_{sol}$), and the injection of excess air arising from the dissolution of entrapped soil air bubbles ($\varepsilon_{EA}$):

$$\delta_{diss} = \varepsilon_{UZ} + \varepsilon_{sol} + \varepsilon_{EA} \qquad (1)$$

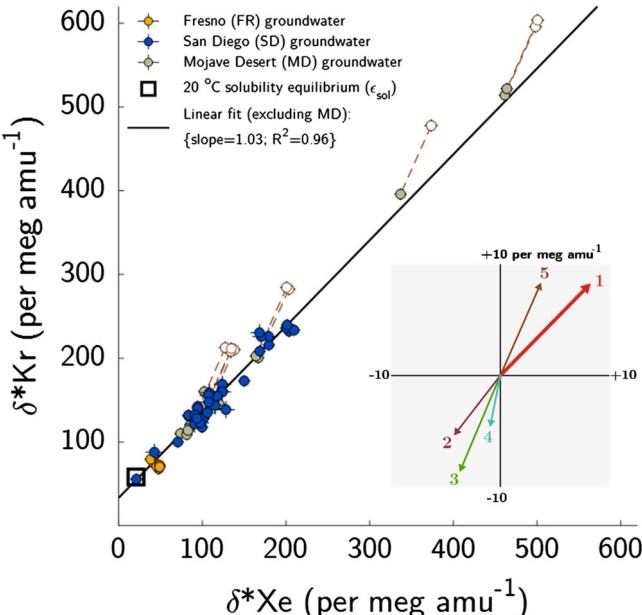

**Fig. 2 Dissolved Kr and Xe isotope ratios in groundwater.** Isotope ratio measurements in 58 groundwater samples collected from 36 wells illustrate the dominance of gravitational settling in driving isotopic departures from solubility equilibrium. Mojave Desert samples for which $\delta^*Kr$ exceeds 200 per meg amu$^{-1}$ are shown both as original values (open circles) and corrected for fractionation due to oxygen consumption (Supplemental Note 5). Error bars indicate ±2-σ uncertainty. Inset: predicted fractionation associated with (1) two meters of gravitational settling; (2) thermal diffusion caused by a 2 °C difference between the surface and water table (WT); (3) a steady-state WT-to-atmosphere water-vapor flux driven by a 0.81% WT-atmosphere absolute humidity difference (equivalent to 20 °C, 65% relative humidity surface air); (4) complete dissolution of entrapped soil air bubbles (equivalent to 50% ΔNe); and (5) steady-state oxygen depletion leading to a 0.5% atmosphere-to-deep unsaturated zone difference in the sum of $O_2$ and $CO_2$. Note that the inset scale is magnified.

Solubility fractionation refers to the preferential dissolution of heavy Kr and Xe isotopes, versus light isotopes, in fresh water at solubility equilibrium[6]. The isotopic influence of excess air opposes that of solubility fractionation by an amount that depends on the quantity of initially entrapped soil air and the completeness of its dissolution under the assumption of a closed water-bubble system at solubility equilibrium[2]. A recent study of solubility and kinetic fractionation of Kr and Xe isotopes in fresh water demonstrated that $\varepsilon_{EA}$ is negligible (order 1 per meg amu$^{-1}$) and $\varepsilon_{sol}$ is largely insensitive to temperature[6]. In principle, therefore, differences between dissolved Kr and Xe isotope ratios in groundwater and atmospheric air should primarily arise from only two processes: gravitational settling, which is the primary control on $\varepsilon_{UZ}$, and solubility fractionation (Fig. 1).

**High-precision Kr and Xe isotope measurements in groundwater.** To test this expectation, we compare the mass difference-normalized, error-weighted means of Kr and Xe heavy-to-light isotope ratios ($\delta^*Kr$ and $\delta^*Xe$, respectively) in 58 groundwater samples collected from three regions in California: Fresno, the Mojave Desert, and San Diego (Fig. 2). Because gravitational settling depends only on isotopic mass difference, it leads to an identical increase in these mass difference-normalized isotope

ratios with depth (i.e. $\delta^*Kr$ and $\delta^*Xe$ will covary with a slope of 1 if fractionated only by gravitational settling).

Indeed, we find that $\delta^*Kr$ and $\delta^*Xe$ in all samples are greater than or equal to $\varepsilon_{sol}$, and a linear regression through all measurements has a slope of $1.16 \pm 0.02$ ($R^2 = 0.98$). The 40 Fresno and San Diego samples fall along a linear trendline with a slope of $1.03 \pm 0.04$ that originates at the solubility equilibrium value. As described above, other non-gravitational isotopic fractionation processes in UZ air and groundwater generally exhibit different $\delta^*Kr$ vs. $\delta^*Xe$ slopes from gravitational settling and are both opposite in sign and smaller in magnitude. For context, as little as two meters equivalent of gravitational settling fractionation in UZ air exceeds the expected individual magnitudes of Kr and Xe isotopic fractionation associated with each of the following: thermal diffusion driven by a 2 °C water-table-to-surface temperature difference, a steady-state water-vapor flux from the moist UZ into 20 °C, 65% relative humidity surface air, and complete dissolution of entrapped soil air bubbles yielding excess air equivalent to 50% Ne supersaturation (ΔNe; Fig. 2).

Mojave Desert groundwater samples exhibit slight departures from the expected concordant Kr and Xe relationship that are most apparent in samples collected from regions of deep present-day water table depths (>50 m). We suspect that an additional physical process systematically affects these deep samples. Two candidate mechanisms are disequilibrium kinetic fractionation[18] driven by barometric pumping or oxygen consumption[22,23] without equimolar replacement by $CO_2$ in deep UZ air (Supplementary Note 5). We explored a simple model for the latter process, in which the mole fractions of $O_2$ and $CO_2$ in deep UZ air above the water table together comprise ~16% of dry air. This is equivalent to a 5% depletion from the overlying atmosphere, in which $O_2$ and $CO_2$ comprise ~21% of dry air. This 5% depletion falls within the observed range of $O_2$ consumption in past UZ air studies[22,23] and may explain the discordant Mojave Desert data, as illustrated in Fig. 2.

**Inferring WTD from dissolved $\delta^*Kr$ and $\delta^*Xe$.** The close agreement of groundwater $\delta^*Kr$ and $\delta^*Xe$ with the expected gravitational slope is encouraging for quantitative reconstruction of past WTD based on isotopic measurements. To explore this possible application, we developed an inverse model that estimates past WTD, recharge temperature, and excess air parameters[2] constrained by measured noble gas isotope ratios and elemental concentrations. With knowledge of both recharge temperature and WTD, it is now possible to quantify and remove the contributions of UZ fractionation and geothermal heat to reconstructions of past surface temperature, thereby resolving two longstanding concerns about noble gas paleo-thermometry[20]. Our inverse model approach merges existing models for equilibrium dissolution and excess air[2] and UZ air fractionation[13] in an iterative loop that converges on best estimates of WTD and temperature (Supplementary Fig. 2). Analytical measurement uncertainties are propagated to estimate probability distributions of WTD and temperature via Monte Carlo simulations.

We first tested this inverse model with five samples collected from two Fresno wells of relatively young recharge age, ~650 and ~35 years, which were constrained by multiple independent dating tools ($^3H/^3He$, SF$_6$, and $^{85}Kr$, $^{14}C$, and $^{39}Ar$). Using measured Ar, Kr, and Xe isotope ratios and elemental concentrations, we reconstructed WTDs for comparison with nearby historical water-level observations. The younger samples, which have a mean recharge year of 1983 CE (±5 years), agree closely with the mean of nearby water-level observations in the early 1980s (Fig. 3). The older samples fall within the range of decadal variability prior

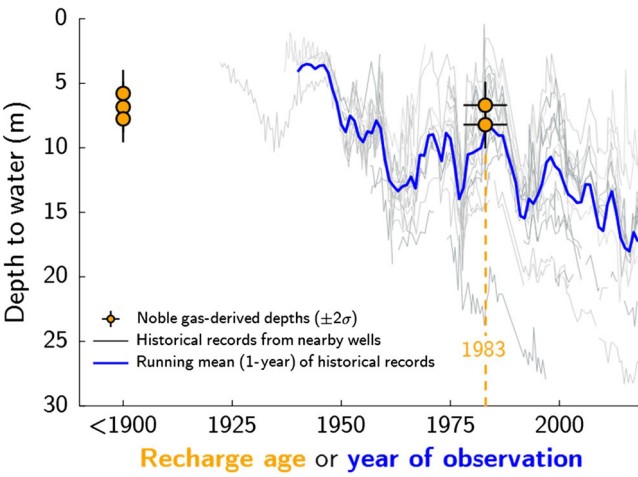

**Fig. 3 Modern validation of noble gas isotope-derived water table depths.** Water-table depth estimates derived from groundwater noble gas isotope measurements are compared to historical water-level observations near two adjacent wells sampled in March 2018 near Fresno, California. Depths are relative to the land surface. Historical records from within a 10-km radius of the sample site were included, except for those from the highly populated city center, which dropped substantially in the mid-20th century (Supplementary Fig. 4). Multiple age tracers were used to determine probable recharge ages (Supplementary Fig. 5).

to 1950, before major development of the Fresno region, and likely reflect mean mid-to-late Holocene regional WTD due to smoothing of isotopic signals by dispersive mixing (Supplementary Note 3).

**Reconstructing past hydroclimatic shifts in WTD.** To investigate the potential for WTD reconstruction as a novel tool for paleoclimate, we collected and analyzed samples from 23 San Diego monitoring wells, 18 of which contain groundwater from a confined regional aquifer system with [14]C-dated recharge ages between 5 ka and 40 ka (Supplementary Note 4). Here we consider noble gas isotopes and concentrations in samples collected from these 18 monitoring wells, which individually provide access to groundwater at various depths at six separate locations in the western portion of the recharge area (Fig. 4 inset). The regional groundwater system flows from east to west and is presumed to recharge over high-permeability alluvial deposits near stream channels west of the mountainous topography and crystalline surficial geology that characterize the region 10–15 km east of the well sites (Supplementary Note 4). Fundamentally, paleoclimate reconstruction using groundwater relies on an understanding that dispersive intra-aquifer mixing of groundwater originating from various times and places of recharge smooths climatic signals and acts as an effective spatiotemporal low-pass filter[20]. Therefore, we primarily expect low-frequency climatic signals to be preserved by noble gases in groundwater, which thus provide high quality information about long-term changes in mean regional WTD and temperature across a major climate transition such as the last deglaciation.

As shown in Fig. 4, Kr and Xe isotope-based reconstructions reveal considerably shallower WTDs throughout the latter portion of last glacial period (LLGP; defined here as 15–40 ka) before deepening during the last deglaciation (Fig. 4). Excluding samples from two outlier wells discussed below, we find mean regional WTDs during the LLGP and post-LLGP of 21.1 ± 0.8 m (±1 SE; 20 samples from 11 wells) and 39.1 ± 1.0 m (±1 SE;

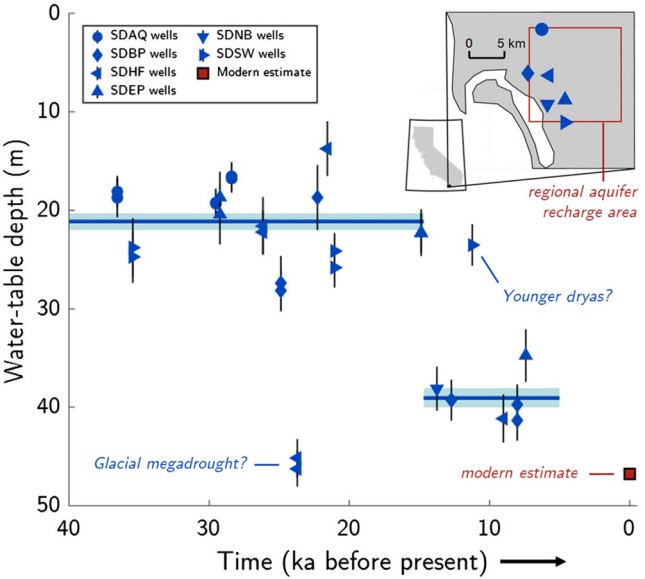

**Fig. 4 Reconstructed water-table depths in the San Diego region.** Noble gas isotope-derived water-table depths (WTD) from the San Diego aquifer system are presented in meters below the land surface. Blue markers and error bars (±2σ) indicate WTD derived from individual samples collected from wells at multiple depths at six sites (indicated by marker shapes). Pre and post 15-ka mean WTDs (lines with ±2-SE shaded regions) exclude two extreme WTD observations that could reflect a previously documented glacial dry period and wet Younger Dryas stadial. The present-day mean WTD over the assumed recharge area (inset) was estimated by a high-resolution, observation-constrained groundwater model (red square marker). Groundwater age uncertainties are roughly ±2 ka (Supplementary Note 4).

6 samples from 5 wells), respectively, indicating a deglacial WTD decline of 17.9 ± 1.3 m (±1 SE). Our finding of a shallower mean WTD during the LLGP is consistent with previous paleoclimate proxy reconstructions and model simulations[7–11], which have suggested that glacial-period wintertime storm tracks were displaced southward due to interaction of the mean atmospheric circulation over western North America with the Laurentide Ice Sheet (LIS), delivering enhanced rainfall to the southwestern United States. A decline in WTD during the last glacial termination is indicative of a shift towards drier conditions, possibly linked to a northward migration of the wintertime storm tracks in response to the receding LIS. Although smoothing of abrupt climate signals in groundwater by dispersive mixing[20] precludes placing tight constraints on the timing of the deglacial WTD shift, the apparent timing of roughly 15 ka is temporally consistent with the rapid lowering of the LIS[24]. Simulation of mean WTD over an assumed recharge area by a high-resolution groundwater model[25] indicates a present-day regional-mean WTD of 47 m, which is substantially below the 21 m LLGP-mean WTD. Although topographic variability leads to spatial inhomogeneity in WTD, we suggest that dispersive mixing integrates isotopic WTD signals from water recharged over a wide spatial range and may therefore damp variability arising from spatial differences in topography. The close agreement of samples from 11 of 12 total LLGP wells supports the notion that the observed change in WTD reflects a regionally coherent groundwater response to a major hydroclimatic shift.

For the first time, we are able to reconstruct surface temperature by accounting for both geothermal heat and UZ air fractionation in the San Diego regional aquifer samples. We find that the mean temperature during the LLGP was 13.6 ± 0.2 °C (±1 SE) before

warming to $19.9 \pm 0.5\,°C$ ($\pm 1$ SE) in the Holocene, which closely matches modern regional surface temperatures (Supplementary Fig. 8). Two San Diego multiple-depth well sites exhibit high apparent LLGP recharge temperatures, which may have a geothermal origin or may indicate adsorption (Supplementary Fig. 9) and are thus excluded from our surface temperature analysis. We demonstrate that any effect on Kr and Xe isotopes in these two samples, and therefore on WTD estimation, is negligible (Supplementary Note 4, Supplementary Fig. 10).

Two wells with outlier reconstructed WTDs have apparent [14]C-based recharge ages of ~24 ka and 12 ka, which may be temporally consistent with other studies that have proposed evidence for a local glacial megadrought and wet Younger Dryas stadial, respectively. Measurements of various physical, geochemical, and biological proxies from Lake Elsinore, roughly 100 km north of our SD study area, independently suggest extreme aridity persisting from ~27.5 to 25.5 ka[26–28], which is within the dating uncertainty of the ~24 ka groundwater sample. A record of groundwater-discharge deposits from southeastern Arizona suggests the persistence of a shallow, near-surface regional water table from ~50 ka to 15 ka, after which the water-table lowered, except for a brief rebound to wet conditions during the Younger Dryas stadial ~12.5 ka[29]. Although reconstructed WTDs from these San Diego wells in principle support the notions of regional dry and wet periods around 24 ka and 12 ka, respectively, we cannot be certain that these reconstructed WTDs represent climatic signals rather than anomalous hydrogeological conditions. Similarly, the preservation of climatic signals from these periods would require minimal dispersive mixing along the flow path, which we cannot conclude based on our limited knowledge of the groundwater flow system.

**Future outlook on water-table depth reconstruction.** Based on the observed concordance of Kr and Xe isotopic measurements in groundwater with the expected primary influence of depth-dependent gravitational signals, we suggest that these measurements represent a promising new tool for hydrogeology and paleoclimatology. By analyzing dissolved Kr and Xe isotope ratios at high precision for the first time, we have (a) confirmed the dominant role of UZ gravitational settling in setting dissolved Kr and Xe isotope ratios in groundwater, (b) found close agreement between observed and reconstructed WTDs in Fresno samples of recent recharge, and (c) quantified a pronounced decrease in San Diego regional WTD during the last deglaciation. While future experiments may shed light on small departures from gravitational expectation as seen in the Mojave Desert samples, our findings offer strong support for the notion that heavy noble gas isotopes in groundwater are quantitative recorders of WTD at the time of recharge. Knowledge of past WTD not only provides an important correction to groundwater noble gas paleo-temperature reconstruction, but also adds complementary hydroclimate information. Future applications to constrain regional-scale groundwater flow models may enable quantitative reconstruction of past precipitation-minus-evaporation rates and improve our understanding of groundwater hydrogeology in aquifer systems presently supporting large populations.

## Methods

**Groundwater sampling.** In this study, a total of 58 groundwater samples from 36 wells were collected in evacuated two-liter flasks and analyzed in the Scripps Institution of Oceanography (SIO) Noble Gas Isotope Laboratory. The borosilicate glass flasks used in this study were made to include two necks leading to 9-mm diameter Louwers-Hapert valves with double o-ring inner and outer seals. Sampling flasks were prepared, stored, and filled following procedures described in Seltzer et al.[6], which are based on seawater dissolved gas sampling techniques[30]. Before and after sample collection, the necks and cavities between the inner two

o-rings were flushed with nitrogen gas before the cavities were sealed and necks closed off with rubber caps. All samples were collected within 5 days of evacuation and analyzed within 2 weeks of collection. A comparison of replicate sample differences in $\delta^{40}/_{36}$Ar across 1 year of distilled water, groundwater, and seawater sample analyses, including those in this study, revealed no sensitivity to storage time either before or after sampling, confirming that any fractionation due to permeation of noble gases across the double o-ring seals and through the $N_2$ filled necks and cavities was below analytical detection[6].

In the field, three well casing volumes were purged from each well prior to sampling. Flasks were first prepared for sampling by attaching 1.25-cm inner-diameter Tygon tubing to the outer neck of the flask and flushing the tubing up to the inner o-ring seal with $N_2$ gas flowing through 0.3-cm Nylon tubing inserted into the Tygon tubing. With the $N_2$ gas flowing, 0.6-cm outer diameter Nylon tubing was connected to the groundwater pump and inserted into the Tygon tubing up to the inner o-ring seal. The 0.3-cm diameter tubing was then removed and the Tygon tubing was flushed with the groundwater, with care taken to dislodge any bubbles. At this point, sampling began by cracking open the Louwers-Hapert valve to allow groundwater to enter the evacuated flask while maintaining a buffer volume of groundwater in the neck and Tygon tubing to prevent any incorporation of atmospheric air. Once the flask was ~95% full, the Louwers-Hapert valve was closed and the cavity between the o-rings as well as the neck volume were filled again with $N_2$ gas and capped.

**Extraction and purification of dissolved gases.** At SIO, samples were weighed both before and after sampling for the purpose of determining bulk dissolved gas concentrations via manometry. Dissolved gases were quantitatively extracted from each sample by sparging with gaseous helium and cryogenically trapping liberated gases in a diptube immersed in a 4-K liquid helium dewar[6]. After a sample was initially connected via 0.5″ ultra-torr fittings to the extraction line, the space between the o-ring seal and vacuum line connection was evacuated to <0.1 mTorr and leak checked. Next, the sample was poured into a 6-L extraction vessel, which was initially under vacuum, and all remaining gases in the two-liter flask were cryogenically trapped over a 15-min period. At this point, the now-evacuated two-liter flask was closed off to the extraction line and 1 atm of ultra-high-purity helium gas was added to the entire line, connected to a flow-through diptube (in liquid helium) and recirculating Metal Bellows MB-41 pump. Over a 90-min extraction period, the helium gas was recirculated at a $1\,L\,min^{-1}$ flow rate and the diptube was progressively lowered into the dewar to prevent saturation of available surface area for gas trapping. After this period, helium gas was pumped away and the diptube was closed off and removed from the liquid helium dewar. Repeated testing of this extraction technique demonstrated that $99.7 \pm 0.1\%$ of dissolved gases were extracted[6]. For each sample, extracted gases were gettered both by SAES Zr/Al sheets and Ti sponge for 130 min at 900 °C to remove all reactive gases. The remaining (noble) gases were then cryogenically transferred into another diptube immersed in liquid helium.

**Isotope-ratio mass spectrometry.** After allowing at least 3 h at room temperature for homogenization, the gases in the diptube were expanded into a calibrated volume attached to the inlet of a Thermo-Finnigan MAT 253 isotope-ratio mass spectrometer and total pressure was measured using a 100-Torr Baratron capacitance manometer. With knowledge of the sample weight, total (noble) gas pressure, volume, and measurement temperature, dissolved Ar concentrations were calculated by making a slight correction in all samples for the presence of Ne (the most abundant of the other noble gases, representing ~0.1% of total noble gas pressure) and gas non-ideality using the second Virial coefficient[31]. The sample was then introduced to the bellows of the mass spectrometer and given 10 min for homogenization before analysis. Over a ~9-h analysis period, isotope ratios of Ar, Kr, and then Xe were measured in sequence with multi-collector Faraday cups, followed by elemental ratios (Kr/Ar and Xe/Ar) via peak jumping. Dissolved isotope and elemental ratios were measured dynamically against a working standard gas[19]. Atmospheric air samples were collected and analyzed against this same working standard to normalize all measured isotope and elemental ratios to the well-mixed atmosphere. Detailed overviews of analytical corrections and uncertainty are provided in Supplementary Note 1.

## Data availability

All data presented in this study are included in this published article as a supplementary data set, which is also publicly available online through PANGAEA (https://doi.org/10.1594/PANGAEA.907908 [https://doi.org/10.1594/PANGAEA.907908]).

## Code availability

The code generated for analyses in this study is available from the corresponding author on request.

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

## Acknowledgements
We thank Ross Beaudette, Sarah Shackleton, Bill Paplawski, Ray Weiss, and Adam Cox for analytical advice, support, and equipment loans, and Greg Mendez, Ernesto Araiza, Kate Durkin, Matt Pendergraft, Robert Kent, and Ray Cordero for assistance in sampling. This work as supported by NSF awards EAR-1702704 (to J.S.), EAR-1702571 (to M.S.), an NSF Graduate Research Fellowship (to A.S.).

## Author contributions
J.P.S., A.M.S., and M.S. developed the underlying concept and designed the research plan. W.R.D. facilitated San Diego groundwater sampling and advised on hydrogeological interpretation. J.T.K. coordinated Fresno groundwater sampling and assisted with data interpretation. A.M.S., J.N., and R.S.G. collected groundwater samples. A.M.S. performed laboratory and data analysis, developed laboratory and inverse model methods, and wrote the original article draft. All authors participated in review and editing of the article.

## Competing interests
The authors declare no competing interests.
