## [Peer Review File · Nature Communications]

Reviewers' comments:

Reviewer #1 (Remarks to the Author):

Review by Werner Aeschbach

Summary

The authors presents a comprehensive study of the potential of high-precision noble gas isotope measurements in groundwater for the reconstruction of past water-table depths. This to my knowledge completely new isotope tool has become available after a recent, highly impressive analytical progress achieved by some of the authors (Seltzer et al., 2019, Ref. 6). The current manuscript now presents the logical follow-up, exploring and demonstrating the use of such measurements to determine groundwater table depths.

The study goes far beyond a first feasibility test, as it includes a large number of samples from three different sites. The data are of exquisite quality and their interpretation reaches a depth that only leading experts in the field could achieve. Indeed, the author group combines leading researchers in noble gas analysis and application to groundwater. The biggest difficulty with the manuscript appears to be the complexity and novelty of the method, leading to the necessity to back up the general story of the main text (that should remain understandable to non-experts) with detailed methodological explanations in the supplement (that peers will need to follow up all the arguments). This is generally well-done, but I have some comments in this context.

In summary, the manuscript definitely deserves publication in a high-level journal such as Nature Communications. It is highly innovative and potentially opens up a completely new field for noble gas studies in groundwater and paleoclimate research. However, some moderate revisions may help to clarify certain points and to make the paper more accessible for a wide range of readers.

Detailed comments

1. Both in the abstract and at the end of the introduction, reference is made to a hydroclimatic shift from a wetter glacial period to a more arid Holocene. I found this astonishing, as generally I would rather expect the opposite. Only later (lines 176 - 180) it becomes clear that this finding is in agreement with other studies and due to a regional shift in storm tracks. I suggest to give the reader a bit more information already at the end of the introduction, e.g. by adding a statement such as "presumably due to a northward shift of storm tracks" or so.

2. It should be made clear in the main text (as opposed to only the supplement) that the calculations of the effect of excess air assume that the entrapped air is soil air, affected by the gravitational settling fractionation, not atmospheric air. This is not explicit in the discussion of equation (1), but otherwise ϵ_{EA} would probably be quite large. It becomes more problematic in the discussion of Fig. 2, when on line 112 the authors speak of "complete dissolution of entrapped air" and in the caption (somewhat imprecise) of "complete dissolution of unfractionated excess air", which seems to suggest that unfractionated atmospheric air is added to the dissolved phase. In my understanding, however, it is gravitationally fractionated soil air from the depth of the water table, which is entrapped and (partly or completely) dissolved. My recommendation is to speak of "soil air" rather than just air, or "entrapped bubbles of soil air", on lines 87 and 112, and to avoid the term "unfractionated" altogether.

3. When discussing the effect of oxygen depletion (which I consider to be quite likely), I find it somewhat unclear to speak of a 0.5 or 5 % difference in the sum of $O_2 + CO_2$, because this could mean a relative change (5 % of the mole fraction of ~ 0.21) or an absolute change in the mole fraction expressed in percent (i.e., from 21 % to 16 %). I don't really have a clear solution to this problem. In the supplement, introducing the change in mole fraction ΔX_{O_2} and giving these values makes it clear, but in the main text it remains ambiguous.

Further confusion may come from first talking of an 0.5 % difference, referring to the inset in Fig. 2, and then of a 5 % effect, referring to the correction shown for the samples. This can be understood as soon as one realizes that the scale of the inset is heavily magnified compared to the scale of the main part of Fig. 2. Maybe it would help readers if this was explicitly stated in the caption of Fig. 2.

4. It is a bit of a pity that the noble gas temperature reconstruction for the SD case study remains quite low-key, with the respective figure hidden in the supplement (Fig. S8). Maybe the difficulties with too high NGTs of some samples, which are discussed mainly in the supplementary Materials (around Fig. S9), which indeed are difficult to explain, are a consideration in this decision. And of course, the focus is on the water table reconstruction, which is a new method and requires detailed explanations.

In general, this manuscript is very dense, with lots of interesting and important information (e.g., about the inverse model for WTD determination, the study areas, the groundwater dating, the NGTs, and special effects/problems seen in some samples) outsourced to the supplement. I understand and appreciate the authors' aim to present a comprehensive study, but I have the impression that there is material for at least two papers in this manuscript. This probably can and should not be changed at this stage, but may be a consideration for how to proceed with the presented and future material. The method (or rather set of methods) is quite complex and may need a few further studies before the community becomes acquainted with it.

5. The main result of the manuscript is the water table shift shown in Fig. 4. While this finding is quite impressive, there is one point that at least from the main text is not sufficiently clear to me. The method reconstructs absolute water table depths (WTDs) below ground surface at the time of recharge. While changes in WTD at one site over time are climatic signals (in the absence of direct human impact), the variability in WTD between different places may mainly be due to local topographic features. This is discussed in section 4 of the supplement and the solution to arrive at representative regional mean WTD appears to lie in using a groundwater model to estimate the modern mean WTD. Somehow, I am not entirely satisfied here. How well do the sampled wells and results from the Holocene and late Pleistocene represent regional mean WTDs? As the sampled wells are clustered in six multi-depth well sites, can the WTD results from each well site but different ages be directly compared? At least, a short consideration of the effect of topographic variability on the result shown in Fig. 4 should be added to the main text.

6. It would be interesting to see how excess air amounts are related to WTD changes, as seen in Fig. 4. Literature about relationships between excess air and WTD (changes) is not mentioned in the main text, only in the Supplementary Materials. Maybe the current data are not sufficient to look at this in more detail, especially since Ne concentrations are missing, but experts in the field may wonder about such links. Maybe a statement could be made in the main text.

With regard to the discussion about possible excess air influences on Ar and Kr concentrations in the Supplementary Materials (discussion of Fig. S9), I wonder whether this question cannot be answered based on the excess air parameters (A, F) obtained from the coupled inverse modeling undertaken in this study. Certainly, additional Ne concentrations would help to pin this down, but in principle the available data may suffice to constrain the excess air. As far as I can see, Ne was not available for the SD samples discussed here. However, in the Fresno case study, possibly Ne was measured in the context of the ^3H - ^3He dating? Would including this information not improve the parameter estimations?

By the way, there are several more papers that address the relationship between excess air and water table fluctuations / humidity, summarized in section 4.2 of the review by Aeschbach-Hertig and Solomon, 2013 (Noble gas thermometry in groundwater hydrology. In: Burnard, P. (Ed.), *The Noble Gases as Geochemical Tracers*. Springer, pp.81-122).

7. Supplementary Materials, equations (S4) and (S5): I don't understand why the mass differences for the different isotope ratios do not enter in these equations. Comparing with the equations (3) and (4) in Seltzer et al., 2019, there seem to be corresponding factors in the

denominator terms missing.

Minor corrections

Line 65: "Isotope ratios of ... **are** more strongly affected by ...".

Line 181: "... indicative of **a** shift ..."

Line 214 (caption Fig. 4): Delete "be"

Line 260: evacuat**ed**

Supplementary Materials, line before eq. (S7): "Extension **of** S6 ..."

Supplementary Materials, bottom of p. 8: "Piston flow age distributions f**ro**m each age tracer ...".

Supplementary Materials, top line of p. 9: Delete "Kip".

Supplementary Materials, caption of Fig. S11: "brin**g**ing several uncorrected values ...".

Reviewer #2 (Remarks to the Author):

Please see attached files for the recommendation to the editors and general comments on the manuscript as well as the minor comments in form of annotated comments in the original pdf files of the Main Text and Supplementary Materials.

Reviewer #1 (Werner Aeschbach)

Detailed comments

1. Both in the abstract and at the end of the introduction, reference is made to a hydroclimatic shift from a wetter glacial period to a more arid Holocene. I found this astonishing, as generally I would rather expect the opposite. Only later (lines 176 - 180) it becomes clear that this finding is in agreement with other studies and due to a regional shift in storm tracks. I suggest to give the reader a bit more information already at the end of the introduction, e.g. by adding a statement such as “presumably due to a northward shift of storm tracks” or so.

We agree that this point – agreement with the direction of paleo-hydrological change identified by prior studies – should be made early on. We have added a sentence to this effect in the second paragraph of the main text.

2. It should be made clear in the main text (as opposed to only the supplement) that the calculations of the effect of excess air assume that the entrapped air is soil air, affected by the gravitational settling fractionation, not atmospheric air. This is not explicit in the discussion of equation (1), but otherwise eps_EA would probably be quite large. It becomes more problematic in the discussion of Fig. 2, when on line 112 the authors speak of “complete dissolution of entrapped air” and in the caption (somewhat imprecise) of “complete dissolution of unfractionated excess air”, which seems to suggest that unfractionated atmospheric air is added to the dissolved phase. In my understanding, however, it is gravitationally fractionated soil air from the depth of the water table, which is entrapped and (partly or completely) dissolved. My recommendation is to speak of “soil air” rather than just air, or “entrapped bubbles of soil air”, on lines 87 and 112, and to avoid the term “unfractionated” altogether.

This is an important point that we agree requires clarification. We have adopted Dr. Aeschbach’s suggestions in our revised manuscript and clarified that entrapped bubbles contain (gravitationally fractionated) soil air, not atmospheric air. We have updated the caption to Fig. 2 as well as the discussion of excess air in the main text (lines 87 and 112). We have also further clarified that injection of excess air tends to counteract the effect of solubility fractionation.

3. When discussing the effect of oxygen depletion (which I consider to be quite likely), I find it somewhat unclear to speak of a 0.5 or 5 % difference in the sum of $\text{O}_2 + \text{CO}_2$, because this could mean a relative change (5 % of the mole fraction of ~ 0.21) or an absolute change in the mole fraction expressed in percent (i.e., from 21 % to 16 %). I don’t really have a clear solution to this problem. In the supplement, introducing the change in mole fraction DX_O_2 and giving these values makes it clear, but in the main text it remains ambiguous.

Further confusion may come from first talking of an 0.5 % difference, referring to the inset in Fig. 2, and then of a 5 % effect, referring to the correction shown for the samples. This can be understood as soon as one realizes that the scale of the inset is heavily magnified compared to the scale of the main part of Fig. 2. Maybe it would help readers if this was explicitly stated in the caption of Fig. 2.

We thank Dr. Aeschbach for pointing out this potential source of confusion, and we have addressed it in the revised manuscript by clarifying that the simple model demonstrated in Fig. 2 represents a deep UZ composition in which O₂ and CO₂ together comprise 16% of dry air in the deep UZ. We then state that this is a 5% depletion relative to overlying atmospheric air, in which they together comprise 21% of dry air. We hope this will alleviate any potential confusion regarding relative vs absolute depletion of O₂ and CO₂ mole fractions. We have also taken Dr. Aeschbach's suggestion of adding a clarifying note to the caption of Fig. 2 explaining that the inset has a magnified scale.

4. It is a bit of a pity that the noble gas temperature reconstruction for the SD case study remains quite low-key, with the respective figure hidden in the supplement (Fig. S8). Maybe the difficulties with too high NGTs of some samples, which are discussed mainly in the supplementary Materials (around Fig. S9), which indeed are difficult to explain, are a consideration in this decision. And of course, the focus is on the water table reconstruction, which is a new method and requires detailed explanations.

In general, this manuscript is very dense, with lots of interesting and important information (e.g., about the inverse model for WTD determination, the study areas, the groundwater dating, the NGTs, and special effects/problems seen in some samples) outsourced to the supplement. I understand and appreciate the authors' aim to present a comprehensive study, but I have the impression that there is material for at least two papers in this manuscript. This probably can and should not be changed at this stage, but may be a consideration for how to proceed with the presented and future material. The method (or rather set of methods) is quite complex and may need a few further studies before the community becomes acquainted with it.

As Dr. Aeschbach notes, the focus of this paper is a high-level overview of a new proxy for past water-table depth. We decided that it was important to provide all relevant information and related work in the supplement that we felt could not reasonably be decoupled from the water-table depth application, including the noble gas temperature reconstruction. Unfortunately, we agree that the supplement is quite dense, but we suggest that it is worthwhile to make all relevant information available to readers. A focused interpretation of the noble gas temperature reconstruction is in fact part of an ongoing project led by several co-authors that will result in a separate manuscript in the future.

5. The main result of the manuscript is the water table shift shown in Fig. 4. While this finding is quite impressive, there is one point that at least from the main text is not sufficiently clear to me. The method reconstructs absolute water table depths (WTDs) below ground surface at the time of recharge. While changes in WTD at one site over time are climatic signals (in the absence of direct human impact), the variability in WTD between different places may mainly be due to local topographic features. This is discussed in section 4 of the supplement and the solution to arrive at representative regional mean WTD appears to lie in using a groundwater model to estimate the modern mean WTD. Somehow, I am not entirely satisfied here. How well do the sampled wells and results from the Holocene and late Pleistocene represent regional mean WTDs? As the sampled wells are clustered in six multi-depth well sites, can the WTD results from each well site but different ages be directly compared? At least, a short consideration of the effect of topographic variability on the result shown in Fig. 4 should be added to the main text.

We thank Dr. Aeschbach for calling attention to this important point, which was originally addressed in the supplement but not in the main text. In our revised manuscript (in the section entitled “Resolving past hydroclimate change with water-table depth reconstruction”) we have added several sentences of discussion to clarify several key points related to smoothing of spatial WTD variability and preservation of climatic signals in groundwater. Specifically, this section of the main text now directly describes dispersive mixing in groundwater as a spatiotemporal low pass filter and mentions that topographic variability results in spatial WTD inhomogeneity. The revised text now points out that the close agreement of reconstructed WTD from 11 of 12 total LLGP wells from different locations supports the conceptual expectation that mixing acts to smooth signals over a wide spatial range. We therefore interpret the mean reconstructed WTD from these wells as representative of a regionally coherent signal. At the beginning of these section of the main text, we also added several sentences describing the hydrogeological setting and the relative locations of the wells.

6. It would be interesting to see how excess air amounts are related to WTD changes, as seen in Fig. 4. Literature about relationships between excess air and WTD (changes) is not mentioned in the main text, only in the Supplementary Materials. Maybe the current data are not sufficient to look at this in more detail, especially since Ne concentrations are missing, but experts in the field may wonder about such links. Maybe a statement could be made in the main text.

With regard to the discussion about possible excess air influences on Ar and Kr concentrations in the Supplementary Materials (discussion of Fig. S9), I wonder whether this question cannot be answered based on the excess air parameters (A, F) obtained from the coupled inverse modeling undertaken in this study. Certainly, additional Ne concentrations would help to pin this down, but in principle the available data may suffice to constrain the excess air. As far as I can see, Ne was not available for the SD samples discussed here. However, in the Fresno case study, possibly Ne was measured in the context of the 3H-3He dating? Would including this information not improve the parameter estimations?

By the way, there are several more papers that address the relationship between excess air and water table fluctuations / humidity, summarized in section 4.2 of the review by Aeschbach-Hertig and Solomon, 2013 (Noble gas thermometry in groundwater hydrology. In: Burnard, P. (Ed.), The Noble Gases as Geochemical Tracers. Springer, pp.81-122).

We agree that adding Ne measurements in the future will be valuable for relating reconstructed WTD to excess air and evaluating the relationship between these two variables considered by the references mentioned in Dr. Aeschbach’s 2013 review paper. Although we can calculate ΔNe based on T, A, and F for each sample and there is a general weak inverse correlation between ΔNe and WTD, we suggest that a more complete discussion of excess air and its relationship to WTD is best left for a future study, as this manuscript is already quite complex. Regarding the point about evaluating A and F in the samples with low Kr and Xe concentrations (SDHF and SDBP glacial period samples), evaluation of the CE model results in unphysical values for all parameters - T, A, and F. Specifically, A values are too high and T is too warm, so there appears to be some compensation in the model since, for constant F, noble gas concentrations will increase with A and decrease with T. Perhaps this provides some support for the adsorption hypothesis instead of the geothermal heat hypothesis, since the latter should still be physically compatible with the CE model whereas the former should not (simply put, adsorption is not

included in the CE model, but temperature is). However, this line of reasoning is quite speculative and we would prefer to refrain from making any strong conclusions about the source of the observed deficit of Kr and Xe in these SDHF and SDBP glacial samples. Prior USGS analyses did measure Ne in the Fresno samples and the reconstructed temperature was in close agreement with that found by our measurements. Given the weak sensitivity of noble gas isotopes to recharge temperature and excess air, we felt it was best not to add more detail to an already dense supplement to incorporate these measurements, since the impact of any small changes in temperature and excess air is negligible on reconstructed WTDs.

7. Supplementary Materials, equations (S4) and (S5): I don't understand why the mass differences for the different isotope ratios do not enter in these equations. Comparing with the equations (3) and (4) in Seltzer et al., 2019, there seem to be corresponding factors in the denominator terms missing.

The mass differences were inadvertently left out when typing up this equation in the original supplementary text. We thank Dr. Aeschbach and reviewer #2 for noticing this, and we have corrected these equations in the revised text.

Minor corrections

Line 65: "Isotope ratios of ... **are** more strongly affected by ...".

Line 181: "... indicative of **a** shift ..."

Line 214 (caption Fig. 4): Delete "be"

Line 260: *evacuat*ed**

Supplementary Materials, line before eq. (S7): "Extension **of** S6 ..."

Supplementary Materials, bottom of p. 8: "Piston flow age distributions *f*ro**m each age tracer ...".

Supplementary Materials, top line of p. 9: Delete "Kip".

Supplementary Materials, caption of Fig. S11: "*brin*g*ing* several uncorrected values ...".

We have made these changes. Thanks for calling our attention to these typos.

Reviewer #2

Summary

Overall, A. Seltzer and colleagues present a well-thought-out and well-written study based on high-precision analysis of noble gases in ground water samples. The analytical techniques employed are truly cutting-edge, state-of-the-art and, to my knowledge, novel. Seltzer and co-

workers clearly demonstrate that isotopic fractionation effects on Kr and Xe gas, which are caused mainly by gravitational settling of atmospheric air in the porous voids in the ground above the ground water table (in the unsaturated zone), are recorded by ground water. The depth dependency of the gravitational fractionation can, therefore, be used to interpret past ground water table depths at the time of recharge of the corresponding ground water. For old ground water systems, this gives insight into how ground water levels may have changed in response to climate changes in the region, and Seltzer et al. successfully apply the principle to reconstruct ground water table depths (WTD) over the period of the last glacial maximum to the present in two out of three Californian ground water systems. They also offer reasons for the difficulties encountered in interpretation of the reconstructed WTDs in the third ground water system. As agreed upon with the associate editor of Nature Communications, this review focusses on the analytical methods employed in the study, and doesn't evaluate the relevance of the developed techniques to the field of ground water hydrology, in particular, how reliably reconstructed WTDs can be interpreted and their usefulness to the field. From an analytical point of view, therefore, the work of Seltzer and colleagues is truly ground breaking and certainly of interest beyond the field of hydrology. The most important comments on the work are presented below, and more minor ones are in the form of annotated comments in the attached pdf-versions of the Main text and Supplementary Materials. My overall recommendation is that the authors should be invited to revise their work before it is ready for publication in Nature Communications.

Comments:

1) In places, the manuscript makes claims that are not supported by the presented data. I would invite the authors to make all of their claims consistent with their data. I estimate that none of their conclusions should be affected by toning claims down a bit. On the other hand, they would thereby significantly increase the faith that readers will have in their results. In particular, the quoted analytical precision of " ≤ 5 ppm/amu" in the abstract isn't supported by Table S1 in the Supplementary Materials. Based on table S1 quoting " ≤ 10 ppm/amu" would be more appropriate. Also, in the Supplementary Materials, near the end of section 2, a statement is made that the small difference in average δ^*Kr_{res} and δ^*Xe_{res} "...may indicate that a minor process is not captured by the model, perhaps causing WTD biases at or below the 1-meter scale." However, if the average δ^*Kr_{res} and δ^*Xe_{res} are indeed significantly different from each other, then this clearly indicates that there is still room for improvement in the model, and the authors should acknowledge this (unless the reason is a numerical artefact due to the reconstruction model itself). Also, the δ^*Kr_{res} and δ^*Xe_{res} values need to be regarded at the per sample level for single sample analysis. Hence, if a residual δ^*Kr of ~ 20 ppm/amu (the largest δ^*Kr_{res} of an SD sample in Figure S3) is completely ascribed to gravitational settling, then a difference in WTD of about 5-6 m is necessary to explain the difference - at lot more than the claimed 1 m! If I'm not mistaken, I would advise the authors to reword this statement for better agreement with their data.

We agree with the reviewer's suggestion to clarify instances of confusion regarding measurement and model uncertainties. Reviewer #2 identifies two specific instances, each of which is addressed below.

- 1) The first instance regards the mention of " ≤ 5 per meg amu⁻¹ precision" in the abstract and second paragraph of the main text. This value is intended to refer to the*

*reproducibility of individual measurements, given by the pooled standard deviations of replicate samples, and not to the overall uncertainty (i.e. the accuracy, to which Table S1 and equation S2 refer). The pooled standard deviation of Kr and Xe isotope measurements (δ^*Kr and δ^*Xe) was ~ 4 per meg amu^{-1} during analytical campaign A and ~ 5 per meg amu^{-1} during campaign B. However, having seen comment #4 below by Reviewer #4, in which he/she raises an important point about improving transparency in the description of our error analysis, we realize that this point was not made sufficiently clear in the original text. To clarify, we have a) added (“ 1σ ”) to the precision reference in the abstract and main text, and b) added further detail in supplementary section 1 to our error analysis, which we address directly in our response to point #4 below, and c) reworded the sentence directly following equations S4 and S5 in supplementary section 1 to define the campaign A and B precisions in direct relation to the “ ≤ 5 per meg amu^{-1} ” value mentioned in the abstract and main text.*

- 2) *The second instance refers to the claim that model-measurement residuals for Fresno and San Diego groundwater samples are equivalent to one meter or less of gravitational settling fractionation. The reviewer is correct in identifying that a 20 per meg amu^{-1} δ^*Kr residual, the largest observed for Fresno/San Diego samples, is equivalent to 5 m of gravitational settling. In supplementary section 1 (before Fig. S3), the claim of San Diego/Fresno residuals amounting to one meter or less of gravitational settling was intended to refer to the mean values (4 per meg amu^{-1} and 2 per meg amu^{-1} for δ^*Kr and δ^*Xe , respectively), but it was ambiguous as originally written. We thank the reviewer for catching this. To be clearer, we have revised this sentence to indicate that uncertainty in the model, given by residuals, may lead to over/underestimation of water table depth by up to several meters.*

2) In spite of my lack of expertise in the field of ground water (or maybe exactly because of it), I would invite the authors to introduce the mechanisms of ground water recharge in their field sites more clearly to non-experts. Living in a country where ground water is very often supplied directly by river water that has experienced gas exchange directly with the atmosphere, the mechanisms of how and why the Californian ground water should adopt isotope signatures of the air in the UZ and preserve them over timescales of thousands of years weren't immediately clear, and only became apparent after reading deep into the Supplementary Materials.

If limited space in the Main Text were a problem for this, then possibly, the section discussing the samples indicating a “glacial megadrought” or the “Younger Dryas” could be shortened and/or moved to the Supplementary Materials.

We agree that the conceptual basis for unsaturated-zone recharge as a prerequisite for the development of gravitational signals and the fundamental concept of preservation of paleoclimate information by groundwater over 1-10 thousand-year timescales was unclear in the original manuscript. Reviewer #1 (Werner Aeschbach) raised a similar point and suggested to introduce the concept of regional shifts in water-table depth in the San Diego region over the last deglaciation early in the text. We have done this by adding a few sentences at the end of the first section. We have also added clarifying text to the paragraph directly following Fig. 1, the schematic of idealized fractionation in the unsaturated zone and during dissolution.

3) I strongly object to the use of the "per myriad" sign “□”, because a generally accepted understanding already exists that it refers to one part in 10'000 and not one part in a million as is desired here. See e.g.: https://en.wikipedia.org/wiki/Basis_point
Starting to use the symbol by a different definition will only cause confusion. Even if the main use of □ seems to be in Finances. So, all instances of this symbol throughout this work would need to be changed to either "per meg amu-1", or if this is too cumbersome, why not also to "ppm amu-1"? Although that is also in violation of the SI-system and would irritate unit purists. See, e.g., here, section 7.10.3:

<https://www.nist.gov/pml/nist-guide-si-chapter-7-rules-and-style-conventions-expressing-values-quantities>

Personally, I would deem the use of "ppm amu-1" to be acceptable in this context.

Also, in order to emphasize the fact that ppm, % and ‰ are used as substitutes for the constant factors 0.000001, 0.01, and 0.001, respectively, I would support keeping the explicit statement that is already in the text, i.e. "1 ppm = 0.000001 = 0.0001% = 0.001 ‰"

Accordingly, the use of terms like "in units of ‰" need to be eliminated from everywhere within the manuscript (e.g. in the sentences following equation S7), because neither %, ‰, nor ppm are units in the SI-sense, they only represent constant factors. But one could say that a value "... is given in ‰."

It should also be said that the decision of the authors to choose the definition of δ according to equation S1 is a wise one, i.e. omitting the factor of 1000. This allows for interchangeable use of %, ‰, or ppm in the sense of constant factors that the quoted numbers are multiplied by – unlike the unfortunate ordinate-axis label in Figure 2 in Seltzer et al. 2019, that indicates that values are given in ‰, when the factor of 1000 has already been included in the definition of δ in that publication.

We were previously unaware of the use of the “□” symbol in financial world as meaning 0.0001 and thank the reviewer for calling this to our attention. We agree that the symbol should be avoided in this manuscript and we have replaced all instances of “□” with the words “per meg” in the text.

4) There are a few mathematical errors in the equations provided that need to be corrected: Firstly, the formula for error propagation (equation S2) cannot be a proper error propagation for the calculations that must have been used during data reduction. Unfortunately, even when studying the referenced method papers (e.g. Seltzer et al. 2019), it is difficult to derive the exact formulas used for, e.g., normalisation of the raw data to standards.

However, almost certainly there were multiplications involved here, and not addition or subtraction. Therefore, a quadratic addition of the absolute errors of the variables involved is incorrect. An improved error propagation should involve quadratic addition of the absolute errors of the dependent variables scaled by the partial derivatives of the final result with respect to the corresponding dependent variable (this is assuming all variables are independent of each other). Also, the division by the square root of N should only apply to σ_{pld} , and not to the other errors as well (unless they are measured more often during the measurement of replicates as well, which at least in the case of SE_{ext} would be in contradiction to the text).

I advise the authors to provide a proper and thorough error propagation calculation and to

include all errors next to the corresponding values in the data file tables, such that at least calculation of the $\delta^*\text{Kr}$ and $\delta^*\text{Xe}$ values can be easily reproduced. Ideally, all of the raw data used for evaluation of the results presented here together with a description of the calculation steps should be archived together and made publicly available (e.g. via a digital object identifier (DOI) provided by an electronic library). This would significantly increase the transparency of the work and would allow others to reproduce and confirm the reliability of the presented results. It could be in analogy to recommendations that Roth et al. (Chem. Geol., 386 (2014), pp. 238-248, 10.1016/j.chemgeo.2014.06.022) made for reporting high-precision Nd isotope ratio analyses by thermal ionization mass spectrometry (TIMS) – another analytical method that has got a track record in producing isotope ratios measurements with relative uncertainties of a few ppm. In this particular case, disclosing all of the raw data would have helped to resolve a conflict between two research groups, where one group claimed to have observed a ^{142}Nd isotope anomaly that the other group didn't. It turned out that an analytical artefact related to the mass spectrometric analysis method was the cause of the discrepancy. Upon correcting one data set for the effect, the observed anomaly almost completely disappeared. With increasing availability of high-precision isotope ratio analysis, making the performed analyses as transparent as reasonably possible for other research groups will, in my opinion, be of increasing importance in the future.

We thank reviewer #2 for calling our attention to this potential source of confusion in the original manuscript, namely that the specific approaches to normalizing and correcting raw data and the associated propagation of uncertainty were unclear. To address this point, we have rewritten the first two paragraphs of supplementary section 1 and added three equations to define raw sample and atmospheric standard measurements relative to the working reference gas. The reviewer is technically correct that a proper error propagation should include division, since the raw sample delta values (vs the working reference gas) are divided by mean atmospheric air values (vs the same reference gas) and so the associated errors in each (sample measurements and atmospheric air measurements) are not additive. Previously, our error propagation equation had made use of the approximate additivity of delta values. In the most extreme case, for the deepest Mojave Desert sample (~3 per mil vs atmospheric air), we calculate that this approximation would overestimate the true error by 1.9 per meg for $^{136}/_{129}\text{Xe}$, equivalent to ~0.3 per meg amu^{-1} . To be most accurate, we have taken the reviewer's suggestion to rewrite this equation using the correct propagation of uncorrelated error for division/multiplication. We have also taken the reviewer's suggestion of providing a detailed description of the correction and normalization as well as the raw data for all groundwater measurements, atmospheric air measurements (used for normalization), and standard aliquot tests (used for correction) as a new spreadsheet within in the data set to be included with our publication. Finally, the reviewer correctly noticed that the division by N in the standard error calculation should only apply to the sample reproducibility (σ_{pld}) term, not to the uncertainties associated with atmospheric normalization and correction which are not reduced by measurement more replicate samples of the same groundwater. We thank the reviewer for catching this mistake, and we have now corrected the equation. We have also rerun 1000 Monte Carlo simulations for all 58 samples by properly accounting for the shared uncertainty between all samples (uncertainty in atmospheric air values and corrections). The reanalysis of our samples with these new Monte Carlo simulations slightly changed the estimated mean shift in

WTD and surface temperature. The text and supplemental data set have been updated accordingly. We have also adopted the reviewer's suggestion to archive our data set by uploading it to PANGAEA and listing the associated DOI in the "Data Availability" statement in the main text.

5) Then there are errors in equations S4 and S5: I don't think it is only a copying mistake from Seltzer et al. 2019 (equations 3 and 4). The formulas for δ^*Kr and δ^*Xe presented there are also wrong, I think. The original formula for δ^*Kr was apparently taken from, Orsin 2013, but also there no detailed derivation can be found.

The obvious way to calculate a mass difference-normalised, error-weighted mean of the three $\delta x/yKr$ values (x is the heavy isotope and y the light one) would be the following:

Because by normalising a $\delta x/yKr$ to its corresponding (exact) mass difference, the related error of the measured $\delta x/yKr$ is scaled accordingly. I have failed to convert the above equation into equation 3 in Seltzer et al. 2019.

I have also tried all meaningful permutations of $\delta x/yKr$ and $\delta x/yXe$ values listed in the data file together with the errors or propagated errors in Table S1 and inserted them into the available formulas, and I cannot reproduce the δ^*Kr and δ^*Xe values quoted in the data file.

I therefore urge the authors to provide more transparency to the values and calculation methods used here, in form of justifications of the formulas used as well as providing the data necessary to replicate the results.

*We thank the reviewer for catching this error and note that reviewer #1 (Werner Aeschbach) also noticed that we had inadvertently left out the mass-difference factors in the equation. We have corrected these δ^*Kr and δ^*Xe definitions in the revised supplement. Regarding the final point, we have now provided more detail in the descriptions of normalization and error analysis.*

Otherwise, as far as can be concluded from the available information, the employed analytical methods are of high quality and appropriate for the conclusions made by the authors.

One very minor addition: The column "elevation" in worksheet "SIO Measurements" in the data file needs a unit adding to it.

Thanks. We have now added the unit (meters).

I hope these comments help with revision of the manuscript.

REVIEWERS' COMMENTS:

Reviewer #1 (Remarks to the Author):

The authors have adequately and carefully responded to the points raised by both reviewers. I have the impression that the manuscript is now in extraordinarily good shape and ready for publication as it stands, except for final technical corrections. I think I found two minor typos in the text, as detailed below.

Line 96: injection of excess air arising (not: airing) from the dissolution of entrapped soil air bubbles

Line 204: we find (delete: a) mean regional WTDs

Werner Aeschbach

Reviewer #2 (Remarks to the Author):

The authors have thoroughly revised their manuscript and - barring two not-note-worthy exceptions - have seriously and sufficiently addressed all points raised in the first round of review.

Next to the four minor points listed below, there are about ten further very minor corrections to be recommended - mainly typing mistakes or missing units. They have been highlighted and commented in the attached pdf files. But all in all and assuming these minor points are addressed, there is nothing that would warrant another round of review and the authors should be congratulated for their excellent work!

Minor points:

1) Main text, Figure 1:

I'm sorry, I'm not sure if this figure still needs improving more or not.

The idea of duplicating the abscissa at the top of the figure is a good one!

Originally, my main reservation was that one of the two arrows indicating fractionation due to water-vapour flux and thermal diffusion is positioned incorrectly (green and bordeaux coloured arrows, respectively): I was under the impression that the origins of these arrows should be the same at depth = 0 m, and then the arrows should diverge with increasing depth instead of converging to the same point at 50 m depth as they currently do. The whole figure gives a bit of a "cheap" impression and I (admittedly) jumped to the conclusion that the two arrows were poorly aligned graphical objects - possibly because a "snap to grid" feature of the drawing software was activated and caused a misplacement of one of the arrows.

However, the authors have ignored this issue in their revision, and reading deeper into Severinghouse et al. GCA 1996, it seems as if the positioning of the arrows could actually be intentional, because of a near-surface effect that off-sets the vapour-flux fractionation before it assumes a roughly constant gradient with increasing depth. The same could apply for fractionation due to thermal diffusion. If so, then the figure would indeed be correct, but maybe the non-expert reader should be alerted to this subtlety.

Also, based on the numbers presented in Severinghouse et al. 1996, which for noble gases do not relate to isotopic fractionation, one could expect the water-vapour flux to have a larger fractionation effect at depth than thermal diffusion. Currently, and if correct, Figure 1 suggests the opposite. It would be good if the authors checked this.

Either way, such excellent work as presented here deserves a proper and correct visualization of the underlying principles in the form of a figure.

2) Equation S5 in Supplementary Materials:

Error calculations are a lot clearer now, thank you!

But formally this equation still is not quite correct, because a mixed quadratic sum of relative and absolute errors does not make any mathematical sense. The relative errors need to be converted back into absolute errors first. The correct formula (under the assumption of independent errors) would be as follows:

$$SE_{tot}^2 = SE_{ext}^2 + SE_{cs}^2 + d \cdot \sqrt{\left(\frac{spld}{\sqrt{N}} \cdot d'smp\right)^2 + (SE_{atm}/d'atm)^2}$$

where d is the Greek delta value defined in equation S4, $d'smp$ is from equation S2, and $d'atm$ is from equation S3.

Taking the absolute values of $d'smp$ and $d'atm$ is not necessary because of the subsequent squaring of the value.

Presumably, this is only an error in the presentation of this equation S5, and the recalculation of all of the data performed by the authors during revision was correct (by some means). Had equation S5 been used as it stands here, then by the definitions in equations S2 and S3 all SE_{tot} would have been increased by a factor of 100 - 1000 or more due to the small values of $d'smp$ and $d'atm$.

3) Equations S6 and S7 in Supplementary Materials

Thank you for correcting these equations.

In principle, the unit "amu" should also be included every time an exact mass difference is used in the equation.

Given that the $d*Kr$ and $d*Xe$ values in the Data Table file have not changed compared to the previous version of this manuscript: was the original data calculated with the current equations S6 and S7? I'm just asking the authors to confirm this - or make corrections, if necessary. Even if a recalculation were necessary, I would be surprised if any of the conclusions changed.

4) Supplementary data set / Sheet "Raw measurement details" / Cell D2:

The authors write: "*PI refers to pressure imbalance given in per mil as $PI = V_{sa}/V_{ref} - 1$, where V_{sa} and V_{ref} are mean sample and reference gas voltages during analysis".

Please clarify what you mean by "sample and reference gas voltages". To the non-expert it is not clear that (presumably) reference is being made to the intensity of the ion beam currents during analysis, the unit of which is not the volt! Also, different disciplines in mass spectrometry follow different conventions when reporting ion beam currents: some always scale to the voltage equivalent over a $1e11$ Ohm resistor, whereas others don't. In my experience, reporting ion beam currents in units of volts is a source of confusion and should be avoided where possible.

Response to Final Comments from Reviewers

Reviewer comments in black; Author responses in red.

Reviewer #1 (Remarks to the Author):

The authors have adequately and carefully responded to the points raised by both reviewers. I have the impression that the manuscript is now in extraordinarily good shape and ready for publication as it stands, except for final technical corrections. I think I found two minor typos in the text, as detailed below.

Line 96: injection of excess air arising (not: airing) from the dissolution of entrapped soil air bubbles

Line 204: we find (delete: a) mean regional WTDs

Thanks for catching these errors. We have fixed them in the revised text.

Reviewer #2 (Remarks to the Author):

The authors have thoroughly revised their manuscript and - barring two not-note-worthy exceptions - have seriously and sufficiently addressed all points raised in the first round of review.

Next to the four minor points listed below, there are about ten further very minor corrections to be recommended - mainly typing mistakes or missing units. They have been highlighted and commented in the attached pdf files. But all in all and assuming these minor points are addressed, there is nothing that would warrant another round of review and the authors should be congratulated for their excellent work!

Minor points:

1) Main text, Figure 1:

I'm sorry, I'm not sure if this figure still needs improving more or not.

The idea of duplicating the abscissa at the top of the figure is a good one!

Originally, my main reservation was that one of the two arrows indicating fractionation due to water-vapour flux and thermal diffusion is positioned incorrectly (green and bordeaux coloured arrows, respectively): I was under the impression that the origins of these arrows should be the same at depth = 0 m, and then the arrows should diverge with increasing depth instead of converging to the same point at 50 m depth as they currently do. The whole figure gives a bit of

a "cheap" impression and I (admittedly) jumped to the conclusion that the two arrows were poorly aligned graphical objects - possibly because a "snap to grid" feature of the drawing software was activated and caused a misplacement of one of the arrows.

However, the authors have ignored this issue in their revision, and reading deeper into Severinghouse et al. GCA 1996, it seems as if the positioning of the arrows could actually be intentional, because of a near-surface effect that off-sets the vapour-flux fractionation before it assumes a roughly constant gradient with increasing depth. The same could apply for fractionation due to thermal diffusion. If so, then the figure would indeed be correct, but maybe the non-expert reader should be alerted to this subtlety.

Also, based on the numbers presented in Severinghouse et al. 1996, which for noble gases do not relate to isotopic fractionation, one could expect the water-vapour flux to have a larger fractionation effect at depth than thermal diffusion. Currently, and if correct, Figure 1 suggests the opposite. It would be good if the authors checked this.

Either way, such excellent work as presented here deserves a proper and correct visualization of the underlying principles in the form of a figure.

Yes, Dr. Aeschbach is correct in noting that the offsets between the water-vapor-flux and thermal-diffusion arrows is intentional and reflects the fact that the majority of the water-vapor-flux fractionation effect occurs between the land surface and shallow wetting front. Below the wetting front, geothermal heat acts only to very slightly increase the water-vapor-flux effect due to its control on saturation vapor pressure. Thus, the water-vapor-flux is very nearly constant from the wetting front downwards. The thermal diffusion arrow, however, varies linearly with depth. The apparent convergence of the two arrows at ~50 m is purely coincidence. We have edited the caption of Fig. 2 to make this clear and reduced the line width of the arrows to make it easier for the reader to see the transition at the wetting front. We have also defined the wetting front to be one meter for this idealized figure, and we've now noted this in the figure caption.

2) Equation S5 in Supplementary Materials:

Error calculations are a lot clearer now, thank you!

But formally this equation still is not quite correct, because a mixed quadratic sum of relative and absolute errors does not make any mathematical sense. The relative errors need to be converted back into absolute errors first. The correct formula (under the assumption of independent errors) would be as follows:

$$SE_{tot}^2 = SE_{ext}^2 + SE_{cs}^2 + d \cdot \sqrt{(\frac{spld}{\sqrt{N}} \cdot d'smp)^2 + (SE_{atm}/d'atm)^2}$$

where d is the Greek delta value defined in equation S4, d'smp is from equation S2, and d'atm is from equation S3.

Taking the absolute values of d'smp and d'atm is not necessary because of the subsequent squaring of the value.

Presumably, this is only an error in the presentation of this equation S5, and the recalculation of all of the data performed by the authors during revision was correct (by some means). Had equation S5 been used as it stands here, then by the definitions in equations S2 and S3 all SE_{tot} would have been increased by a factor of 100 - 1000 or more due to the small values of d'smp

and d'atm.

We thank Dr. Aeschbach for pointing out that we had inadvertently left δ out of equation S5, and indeed it was not quite correct as written. Each instance of δ , δ'_{smp} , and δ'_{atm} should in fact be replaced by $\delta + 1$, $\delta'_{\text{smp}} + 1$ or $\delta'_{\text{atm}} + 1$. This can be seen by writing equation S4 as:

$$\delta + 1 = \frac{\delta'_{\text{smp}} + 1}{\delta'_{\text{atm}} + 1}.$$

Assuming uncorrelated errors, the relative error propagation equation can be written as:

$$\frac{\text{SE}}{|\delta + 1|} = \sqrt{\left(\frac{\sigma_{\text{pld}}}{N(\delta'_{\text{smp}} + 1)}\right)^2 + \left(\frac{\text{SE}_{\text{atm}}}{\delta'_{\text{atm}} + 1}\right)^2}$$

where SE refers to the total error on δ arising from sample reproducibility and atmospheric standard measurements. Thus, the right-hand side of the above equation, multiplied by $|\delta + 1|$ should appear in equation S5. The correctly written version of equation S5 now reads:

$$\text{SE}_{\text{tot}} = \sqrt{\left(\left(\frac{\sigma_{\text{pld}}}{\sqrt{N}(\delta'_{\text{smp}} + 1)}\right)^2 + \left(\frac{\text{SE}_{\text{atm}}}{\delta'_{\text{atm}} + 1}\right)^2\right) \cdot |\delta + 1| + \text{SE}_{\text{CS}}^2 + \text{SE}_{\text{ext}}^2}.$$

In practice, we make use of the very good approximation of additivity of delta values in our estimation of SE_{tot} by using the following (much-simpler) equation:

$$\text{SE}_{\text{tot}} \approx \sqrt{\left(\frac{\sigma_{\text{pld}}}{\sqrt{N}}\right)^2 + \text{SE}_{\text{atm}}^2 + \text{SE}_{\text{CS}}^2 + \text{SE}_{\text{ext}}^2}$$

For the largest observed δ value in this study (3.5 ‰), we find that the difference between these two equations is <0.1 per meg/amu, validating our use of this approximation. In the originally submitted version of the text, we had only included the above approximation (as equation S5), but reviewer #1 raised issue with the fact that the correct error propagation must account for the division involved in calculating δ (i.e. equation S4). In our final revised supplementary text, we have now separated equation S5 into equations S5a and S5b, where S5b is the above formulation in which approximate additivity of δ values is assumed. We have clarified that our calculation of SE_{tot} was done using equation S5b.

3) Equations S6 and S7 in Supplementary Materials

Thank you for correcting these equations.

In principle, the unit "amu" should also be included every time an exact mass difference is used in the equation.

Given that the d*Kr and d*Xe values in the Data Table file have not changed compared to the previous version of this manuscript: was the original data calculated with the current equations S6 and S7? I'm just asking the authors to confirm this - or make corrections, if necessary. Even if a recalculation were necessary, I would be surprised if any of the conclusions changed.

We have now added “amu” to equations S6 and S7. Because SE_{tot} was calculated using equation S5b, both before and after revisions, and the resulting SE_{tot} values were used in calculating error-weighted mean delta values, δ^*Kr and δ^*Xe in the final data set were unchanged. We did however re-run our monte carlo simulations based on the point raised by reviewer #1’s initial review that SE_{atm} , SE_{CS} , and SE_{ext} are shared amongst all samples, whereas σ_{pld} is not. This had a very slight effect on resulting WTD and temperature estimations, but did not change δ^*Kr and δ^*Xe .

4) Supplementary data set / Sheet "Raw measurement details" / Cell D2:

The authors write: "*PI refers to pressure imbalance given in per mil as $PI = V_{sa}/V_{ref}-1$, where V_{sa} and V_{ref} are mean sample and reference gas voltages during analysis".

Please clarify what you mean by "sample and reference gas voltages". To the non-expert it is not clear that (presumably) reference is being made to the intensity of the ion beam currents during analysis, the unit of which is not the volt! Also, different disciplines in mass spectrometry follow different conventions when reporting ion beam currents: some always scale to the voltage equivalent over a $1e11$ Ohm resistor, whereas others don't. In my experience, reporting ion beam currents in units of volts is a source of confusion and should be avoided where possible.

We have now clarified in the data set file that the ratio of voltages between sample-side and reference-side gases is equivalent to the ratio of partial pressures of the gas of interest (hence the “pressure” imbalance correction). We have also clarified that we are referring to relative beam intensities between sample and reference side. We are not reporting absolute beam currents, so there should be no need to scale or deal with any units, since we’re interested in the ratio. The PIS correction is made routinely for dynamic isotope ratio measurements (e.g. Section 3.1 of Severinghaus and Grachev, 2003).